# Prevalence of the Os Supranaviculare: A Systematic Review with Meta-Analysis

**DOI:** 10.3390/jcm14175934

**Published:** 2025-08-22

**Authors:** Maksymilian Osiowski, Aleksander Osiowski, Maciej Preinl, Grzegorz Fibiger, Katarzyna Majka, Barbara Jasiewicz, Dominik Taterra

**Affiliations:** 1Faculty of Medicine, Jagiellonian University Medical College, sw. Anny 12, 31-008 Krakow, Poland; maksymilian.osiowski@student.uj.edu.pl (M.O.); aosiowski1@gmail.com (A.O.); grzegorzfibiger@gmail.com (G.F.); kasiamajka145@gmail.com (K.M.); 2Ortho and Spine Research Group, 34-500 Zakopane, Poland; 3International Evidence-Based Anatomy Working Group, 31-034 Kraków, Poland; 4Department of Anatomy, Jagiellonian University Medical College, 31-008 Kraków, Poland; 5ARTROMED Orthopedic and Rehabilitation Center, 30-059 Kraków, Poland; 6Department of Orthopedics and Rehabilitation, Jagiellonian University Medical College, Balzera 15, 34-500 Zakopane, Poland

**Keywords:** os supranaviculare, accessory ossicle, avulsion fracture, clinical anatomy

## Abstract

**Background/Objectives**: The os supranaviculare (OSSN), also known as os talonaviculare dorsale, astragalo-scaphoid ossicle, or Pirie’s bone, is a small extra bone that was first described in 1921 by A.H. Pirie and is located at the top front part of the navicular bone or talonavicular joint. The knowledge regarding the epidemiology of the OSSN is scarcely established, as its prevalence remains unknown and varies significantly among multiple studies. This meta-analysis aims to clarify and systematically summarize all available data on the characteristics and prevalence of the OSSN. **Methods**: Four major databases (PubMed/Medline, Embase, ScienceDirect, Scopus) were thoroughly searched for studies reporting original data regarding the OSSN up until May 2025. The protocol of this study was pre-registered on PROSPERO (ID: CRD42025638111) and adhered to PRISMA guidelines. To evaluate the between-study heterogeneity, the 95% prediction intervals (95%PI) were calculated; *I*^2^ statistic and Chi^2^ test were also used. The AQUA-tool was used to assess the quality of included studies. **Results**: In total, 13 studies (18,745 feet) qualified for inclusion in the quantitative analysis. The pooled prevalence estimate (PPE) of the OSSN in the general population was found to be 0.88% (95%CI: 0.62–1.24%). The PPE of the OSSN was higher in males (0.87%, 95%CI: 0.58–1.32%) than in females (0.48%, 95%CI: 0.14–1.64%). The os supranaviculare was similarly prevalent in both European (1.04%, 95%CI: 0.55–1.96%) and Asian (0.87%, 95%CI: 0.66–1.13%) populations. **Conclusions**: the os supranaviculare is a very rare anatomical variation that is present in less than one in a hundred feet. Moreover, although usually asymptomatic, it can occasionally be associated with dorsal foot pain or navicular stress fractures. Accurate differentiation from avulsion fractures is essential to avoid unnecessary invasive treatment.

## 1. Introduction

The Os supranaviculare (OSSN), also referred to as *os talonaviculare dorsale*, *astragalo-scaphoid ossicle*, or *Pirie’s bone*, is a small accessory bone situated at the proximal dorsal aspect of the navicular bone or talonavicular joint close to the midpoint [1] (Figure 1). First described in 1921 by A.H. Pirie, this ossicle has since remained a relatively obscure anatomical variant, yet one with notable clinical implications [2]. Accessory ossicles are frequent incidental findings in foot and ankle imaging and are usually asymptomatic [3]. Yet, following trauma or injury, these ossicles can be misinterpreted as avulsion fractures due to their overlapping radiographic appearance and symptoms [4]. Distinguishing between these entities is critical to ensure the best treatment protocol and outcomes, as fractures may necessitate urgent surgical management, whereas symptomatic OSSN is usually managed initially with conservative measures [4,5,6]. Such diagnostic confusion is widespread in emergency settings, where timely and accurate evaluation can prevent unnecessary orthopaedic interventions or surgeries. In addition to posing diagnostic challenges, the presence of the OSSN has been proposed as a predisposing factor for navicular stress fracture (NSF) [5]. This association is particularly relevant in athletes involved in high-impact activities such as sprinting or jumping, where repetitive stress on the midfoot is pronounced. The navicular bone blood supply may be divided into three areas: the dorsal part of the bone, which is supplied by the medial tarsal branch of the dorsalis pedis artery; the medial plantar part supplied by a branch of the posterior tibial artery; and the central part, which lacks proper vasculation. The hypovascular area is prone to stress fractures, delay healing and non-union of the bone [6]. Due to its location, the OSSN may put additional pressure on poorly vasculated parts of the navicular bone and increase the risk of NSF, especially during intense workouts [7]. However, the association between the two bones remains incompletely defined in the literature, being represented mainly by case reports with few higher-level studies. An in-depth understanding of the anatomical features and clinical significance of the OSSN may aid in early recognition and prevention of such injuries.

Since its first description, the OSSN has been referenced by various authors with conflicting prevalence rates and interpretations. The true prevalence remains contentious: existing studies report rates from 0.7 to 3.5%, yet methodological heterogeneity and limited sample sizes undermine the reliability of these estimates [1,8,9]. This meta-analysis aims to clarify and systematically summarize all available data on the prevalence of the OSSN, explore its sex-related and geographical distribution patterns, as well as analyse and discuss clinical considerations relevant to this ossicle. By integrating sparse and heterogeneous reports into a unified framework, our goal was to furnish clinicians with a definitive reference for accurate diagnosis and appropriate management of the OSSN.

## 2. Materials and Methods

The protocol of this study was pre-registered on PROSPERO (ID: CRD42025638111). We strictly adhered to the Preferred Reporting Items for Systematic Reviews and Meta-Analyses (PRISMA) guidelines [10] (Appendix A). Ethical approval and informed consent were not required for this study, as it is a systematic review and meta-analysis of previously published data.

### 2.1. Search Strategy

A systematic search of PubMed/Medline, Embase, ScienceDirect and Scopus was conducted from the inception of each database until May 2025 for articles providing relevant data on OSSN. The following search terms were applied: “os supranaviculare” OR “supranavicular” OR os talonaviculare dorsale OR os talonavicular dorsal OR astragalo scaphoid ossicle OR pirie bone OR “accessory ossicle”. No restrictions on the language or publication date were employed. After obtaining the full text of each article, a manual search of the reference lists was carried out to identify additional eligible studies. If data were unclear or the full text was unavailable, the corresponding author was contacted via email for clarification. For articles published in languages other than English, medical professionals fluent in those languages assisted with translation and data extraction. A detailed description of the search strategy is provided in Appendix A.

### 2.2. Eligibility Criteria

Original research on any topic related to the OSSN was included in this meta-analysis. Therefore, any study that used a suitable study design (retrospective/prospective evaluation of patient samples with cohort/observational/anatomical study design) and reported original epidemiological, anatomical, and clinical data on the OSSN was considered eligible for inclusion. Studies that provided (1) insufficient or missing data, (2) studies involving animals, (3) studies published in inappropriate formats (such as meta-analyses, reviews, case reports, and conference reports), and (4) studies performed on foetuses or embryos were among the exclusion criteria. There were no restrictions for inclusion of the study based on the age criteria, year of publication or the original language of an article. Each blind evaluation of potential eligible research was performed by two independent investigators (A.O. and M.O) under the supervision of a third experienced investigator (D.T.); in case of any disagreement, the authors extensively discussed the article until a consensus was reached.

### 2.3. Study Selection and Data Extraction

After retrieving articles from the databases, duplicates were removed, and the titles and abstracts of the remaining records were screened. Study selection was performed independently by two investigators (A.O. and M.O.). Inter-rater agreement for study selection and bias assessment was calculated using Cohen’s kappa coefficient, which was 96.4% (95%CI: 94.5–98.4) for title/abstract screening and full-text review, and 94.8% (95%CI: 91.9–97.6) for bias assessment, indicating almost perfect agreement. A third investigator (D.T.) verified the extracted data and resolved any discrepancies by making the final decision. For studies meeting all inclusion criteria, the following data were extracted using a pre-defined spreadsheet: name of the first author, year of publication, characteristics of the studied population (country, sex, number of participants), number of identified OSSNs, and diagnostic modality used in the study (X-ray, cadaveric dissection, CT, MRI). When available, anatomical features of the OSSN (shape and dimensions) and associated symptoms were also recorded. All included studies reported the prevalence of OSSN per foot; therefore, patient-based prevalence was not calculated. Mendeley Reference Manager (version 2.93.0; Elsevier Ltd., Amsterdam, The Netherlands) was used during the study selection process.

### 2.4. Quality Assessment

Two independent investigators (A.O. and M.O.) assessed the quality of included studies using the anatomical quality assessment (AQUA) tool, which provides a structured framework for evaluating potential bias [11]. Any discrepancies were addressed by a third investigator (D.T.), who reviewed the data and issued the final judgement. The assessment covered five domains: (1) objective(s) and subject, (2) study design, (3) methodology characterization, (4) descriptive anatomy, and (5) reporting of results. Each domain was rated as having a “low”, “high”, or “unclear” risk of bias. A single negative response within a domain led to its classification as “high” risk, whereas domains with only positive responses were rated as “low” risk. If the available information was insufficient to allow for an assessment, the domain was marked as “unclear”.

### 2.5. Statistical Analysis

All calculations and the generation of forest and Doi plots were performed using MetaXL version 5.3 (EpiGear International Pty Ltd., Wilston, Queensland, Australia). The pooled prevalence estimate (PPE), along with subgroup analyses based on sex, diagnostic modality, geographic location, and population size, were calculated using a random effects model (as the included studies were drawn from heterogeneous populations) and DerSimonian–Laird estimator [12]. The logit transformation was applied to stabilize the variances, as the proportions of several included studies were close to zero; the results were later back-transformed using the inverse of the applied transformation to present the final estimates as prevalence proportions. A pair-wise method was employed to handle missing data. Analyses were conducted only when data were available from at least three studies. Statistical significance between subgroups was assessed by comparing 95% confidence intervals (95%CI); overlap of confidence intervals was interpreted as a non-significant difference. Heterogeneity across studies was evaluated using 95% prediction intervals (95%PI) when data were available from more than five studies, providing insight into the expected range of effect sizes in future research [13]. Additional heterogeneity metrics included the *I*^2^ statistic and Chi^2^ test. A *p*-value < 0.10 in the Chi^2^ test was considered indicative of significant heterogeneity, while *p*-values < 0.05 were used as the threshold for significance in all other tests; all tests were two-tailed [14]. To explore potential sources of heterogeneity, subgroup analyses were repeated based on the aforementioned variables. Sensitivity analysis was also conducted using the leave-one-out method to assess the robustness of the findings, which remained stable across iterations. The primary outcome was examined for small-study effects and possible publication bias using the Doi plot and Luis Furuya-Kanamori (LFK) index [15]. The LFK index was interpreted as follows: absolute values between 0 and 1 = no significant asymmetry (no significant small-study effect); absolute values between 1 and 2 = minor asymmetry (might suggest small-study effect); absolute values greater than 2 = major asymmetry (strongly suggest the presence of small-study impact). In addition, the ideal sample size was calculated to determine whether the size of the study population was sufficient. The following formula was applied: *n* = (Z^2^(P(1 − P)))/d^2^, where *N* is the ideal sample size; Z is the Z statistic for a level of confidence (set as 1.96); P is the expected prevalence (set as 1%); d is the precision (0.005) [16]. Based on these calculations, a subgroup analysis was conducted to compare studies with adequate vs. inadequate sample sizes.

## 3. Results

The process of identification and inclusion of the studies used in this article is presented in Figure 2. The initial database search yielded 2641 records, with an additional 27 studies identified through reference list screening. After the removal of duplicates, 2567 unique records remained. Of these, 2477 were excluded due to irrelevance or lack of pertinent data. Full-text evaluation was performed on 90 articles, of which 13 met the inclusion criteria and were included in our analysis [1,3,17,18,19,20,21,22,23,24,25,26,27].

### 3.1. Study Characteristics

Table 1 summarizes the main characteristics of the 13 studies included in this meta-analysis. Altogether, the analysis encompassed 18,745 feet derived from research conducted in seven different countries: Turkey (three studies, 4083 feet), Germany (three studies, 3644 feet), Japan (two studies, 4909 feet), England (two studies, 1900 feet), Italy (one study, 2155 feet), USA (one study, 1054 feet), and Jordan (one study, 1000 feet). The publication dates of the included studies spanned from 1921 to 2022. All studies utilized X-ray as the diagnostic modality, thereby eliminating variability and reducing the risk of potential diagnostic bias. Notably, all eligible studies reported the prevalence of OSSN per analysed foot rather than per patient, reducing the need for data conversion which also minimized the risk of misinterpretation and bias.

### 3.2. Quality Assessment

The majority of included studies were classified as being at “low” risk of bias across all evaluated domains. Only a small number of studies were categorized as having a “high” risk of bias, primarily in the domain of “Methodology Characterization”, while several were rated as “unclear”. This uncertainty was primarily attributable to inconsistencies in reporting standards, particularly among studies published in the 20th century (Table 2).

### 3.3. General Prevalence

In total, analysis of 13 studies revealed a PPE of the OSSN in the general population of 0.88% (95%CI: 0.62–1.24%; 95%PI: 0.26–2.97%) (Figure 3). Out of 18,745 analysed feet, 179 were found to exhibit the presence of the OSSN (Table 3). Assessment of small-study effects indicated major asymmetry, with an LFK index of −2.15. Only two studies reported data on laterality of the OSSN: Coskun et al. identified the accessory ossicle unilaterally in eight left feet and five right feet, and in three feet the bone was present bilaterally; while Candan et al. reported two cases on the left and four on the right [1,3]. No morphometric analyses could be conducted, as no study provided adequate data on the size or shape of the OSSN. None of the included studies provided sufficient morphometric data (e.g., size, shape) to allow pooled analysis. No study reported symptoms specifically attributable to the presence of the OSSN. Stratified forest plots are provided in Appendix A.

### 3.4. Sex-Specific Prevalence

Data from three studies, comprising a total of 6111 feet (2986 male and 3125 female), enabled analysis of sex-related differences in OSSN prevalence. The PPE of OSSN presence was higher in the male population (0.87%; 95%CI: 0.58–1.32%) than in the female population (0.48%; 95%CI: 0.14–1.64%); the difference was considered not statistically significant (Table 3).

### 3.5. Geographical Distribution

Geographic subgroup analysis was performed based on 12 of the 13 included studies, categorised into two regions: Asia (six studies) and Europe (six studies). The single study originating from North America was excluded from this analysis due to insufficient regional representation. The highest PPE of OSSN presence was observed in Europe (1.04%; 95%CI: 0.55–1.96%; 95%PI: 0.12–8.32%), followed by Asia at 0.87% (95%CI: 0.66–1.13%; 95%PI: 0.47–1.61%); however, the differences did not reach statistical significance (Table 3).

### 3.6. Prevalence Based on Population Size

Based on the ideal sample size calculation, a minimum of 1521 feet was determined to be necessary to ensure sufficient statistical validity. Accordingly, all 13 studies were stratified into two groups based on their sample size. Seven studies, comprising a total of 5891 feet, fell below this threshold and were categorized as having an insufficient sample size. The remaining six studies, encompassing 12,854 feet, met the required minimum and were deemed adequately powered. The subgroup with smaller sample sizes yielded a PPE of 1.02% (95%CI: 0.75–1.39%; 95%PI: 0.54–1.94%), whereas the larger-sample subgroup showed a lower PPE of 0.79% (95%CI: 0.44–1.42%; 95%PI: 0.10–6.02%). The difference was not statistically significant (Table 3).

## 4. Discussion

The primary aim of this meta-analysis was to determine the prevalence of the OSSN and explore its variability with respect to sex, geographic region, and sample size. To the best of the authors’ knowledge, this is the first comprehensive systematic review and meta-analysis focused on this anatomical variation. Our findings indicate that the OSSN is a relatively rare accessory ossicle of the foot when compared to others more commonly documented in the literature. Among the most frequently reported accessory bones are the accessory navicular (12.6%) [4], os trigonum (9.0%) [28], os peroneum (6.6%) [29], and os vesalianum (0.6%) [30], as the data from recent meta-analyses showed. Other accessory bones, such as the os calcaneus secundarius, os intermetatarseum, os supratalare, os talotibiale, os sustentaculi, or os infranaviculare, are far less frequently studied. Based on limited available data, their prevalence is estimated to be 1% or lower, which aligns with our findings for the OSSN [9,26,31].

Our analysis revealed that the OSSN was slightly more prevalent in males compared to females and appeared more frequently in European populations than in Asian, although the differences were not significant. However, due to insufficient data, we were unable to determine the rate of bilateral occurrence for the OSSN. When comparing these findings to other extensively studied accessory ossicles, distinct patterns emerge. For instance, the accessory navicular bone demonstrated higher prevalence in females than males, was bilateral in approximately 50% of cases, and appeared more often in Asian populations than in European or North American [4]. It was also more frequently detected on X-rays than in cadaveric dissections [4]. Similarly, the os trigonum was found more commonly in males and was reported to occur bilaterally in 33% of individuals; its detection was most common in MRI, followed by CT, X-ray, and least in cadaveric studies [28]. Geographically, the os trigonum was most prevalent in Asia, followed by Europe, and least in North America [28]. In contrast, the os peroneum was more common in males, detected more frequently in radiographic studies than in cadaveric analyses, and occurred most often in North America, less in Europe, and least in Asia [29]. The os vesalianum exhibited a different distribution pattern: it was more prevalent in females, and its occurrence was highest in North America, followed by Europe and Asia [30].

In our study of the OSSN, a notable strength was the consistent use of X-ray imaging as the diagnostic modality across all included studies. This methodological uniformity reduced modality-related bias and allowed for reliable comparisons of prevalence across subgroups, which was not achievable in studies of other ossicles. An additional methodological strength was the uniform reporting of OSSN prevalence per analysed foot rather than per patient across all included studies. Although our eligibility criteria did not impose restrictions on the reporting unit, this consistency eliminated the need for conversions or exclusions that might otherwise compromise the integrity of the meta-analysis. Similar inconsistencies posed significant challenges in previous syntheses concerning the os trigonum and the accessory navicular bone [4,28]. Furthermore, due to limited clinical data, we were unable to assess the prevalence of pain associated with the OSSN or determine its potential contribution to symptomatic presentations, underscoring the need for further research in this area. Larger, population-based studies are required to capture the full spectrum of variability of the OSSN.

### 4.1. Clinical Implications, Diagnosis and Treatment

A common characteristic among accessory ossicles is their typically asymptomatic nature, with most cases discovered incidentally [32]. However, symptoms may develop in certain individuals, often triggered by trauma, dislocation, fracture, or pathological changes in adjacent structures [31]. The clinical presentation is usually non-specific and may be incorrectly attributed to the presence of an accessory bone rather than the underlying injury [31,32]. Therefore, a clear and systematic understanding of these anatomical variants is critical for clinicians to accurately distinguish them from fractures and other acute pathologies. Such knowledge not only facilitates appropriate diagnosis and management but also helps prevent unnecessary orthopaedic referrals and potential misdiagnoses [9].

The OSSN has particular relevance in sports medicine, where athletes frequently experience high-impact and repetitive stress on their feet. Notably, the prevalence of the OSSN has been reported to be more than five times higher in male professional soccer players compared to the control group [8]. Patients with a painful OSSN typically present with dorsal midfoot tenderness aggravated by weight-bearing or athletic manoeuvres. The accessory ossicle is best seen on a lateral ankle radiograph [31]. Advanced imaging (CT/MRI) can demonstrate marrow oedema around the ossicle when symptomatic, and clearly show its smooth, corticated border. Finally, a Tc-99m bone scan can also be helpful to determine the symptomatic presence of an accessory bone [33]. Although no standardized guidelines currently exist for the treatment of symptomatic OSSN, the literature consistently supports a conservative approach as the initial line of management. Across reported cases, authors uniformly recommend beginning with activity modification, immobilization of the foot with bandage or tape, non-steroidal anti-inflammatory drugs, and image-guided corticosteroid injection if necessary [34]. When symptoms do not resolve, excision of the ossicle has been performed with success. The earliest published case described a runner whose chronic dorsal foot pain was relieved by surgical removal of the ossicle [35]. Osteosynthesis of the accessory bone to the navicular has also been described recently. Sugimoto et al. report fusing the ossicle with bone graft to restore anatomy in athletes, achieving union and relief of pain, which is a novel alternative to simple excision [9]. In any surgical approach, care must be taken to avoid the deep fibular nerve and dorsalis pedis vessels, which lie near the dorsal navicular.

The principal clinical relevance of the OSSN lies in distinguishing it from cortical avulsion fractures of the navicular or talar head, which typically result from a twisting injury and often occur in middle-aged women wearing high-heeled shoes [31]. Accurate differentiation between these two entities is critical, as certain talar fractures demand prompt, usually surgical, intervention to avert debilitating long-term sequelae [36]. Unlike a well-corticated OSSN, an avulsion fracture appears as a thin, irregular bone flake with incomplete cortication and is often surrounded by soft tissue swelling. In contrast, the OSSN is ovoid and sharply demarcated, has smooth margins and no surrounding oedema (Figure 4). Nonetheless, radiographic differentiation can be challenging without a compatible history of trauma. Another emerging concern is the potential link between the OSSN and NSFs [9]. In one series of 23 NSFs, 22% of patients exhibited an OSSN, suggesting that a congenital dorsal cortical notch may predispose to stress injury of the navicular body [37,38]. Although the pathomechanism remains unproven, it is hypothesized that the presence of this accessory ossicle alters local load distribution, creating a biomechanical vulnerability. Several case reports and series provide additional clinical context: for example, Kim et al. [6] reported a bilateral OSSN with incomplete NSF in a soccer player who was successfully treated by excision and immobilisation; Drexelius et al. [39] described an adolescent lacrosse and football player with a triad of NSF, osteochondral defect, and OSSN who underwent surgical management with full return to sport; and Bayramoğlu et al. [40] highlighted the diagnostic challenge of distinguishing the OSSN from a cortical avulsion fracture in a professional basketball player. Collectively, although published reports are relatively scarce, these cases indicate that the majority of symptomatic OSSN presentations occur in athletic populations, where the ossicle’s biomechanical properties may increase vulnerability to midfoot injury. Clinicians should therefore maintain a high index of suspicion for OSSN-related pathology in athletes presenting with dorsal midfoot pain, as such cases are likely under-recognised in routine practice.

### 4.2. Methodological Considerations and Limitations

Our heterogeneity assessment underscored the considerable variability that persisted despite subgroup stratification. Overall values indicated substantial between-study inconsistency, as inherent in meta-analyses of this kind given the diversity of sampled populations. While heterogeneity was markedly reduced in certain strata, this impact could not be fully mitigated. These findings suggest that unmeasured differences in study populations, imaging protocols, and reporting standards continue to influence the pooled estimates. Accordingly, our prevalence figures should be interpreted with caution, recognizing that true variability may be greater than our summary metrics imply. Additionally, the paucity of data from Australia, South America, and North America further constrains the broader applicability of our findings. Moreover, our assessment of small-study effects suggests the presence of negative bias, as indicated by the Doi plot and LFK index calculations. This suggests that smaller investigations reporting higher prevalence rates may be underrepresented in the literature, whether through publication practices that favour lower prevalence findings, or the exclusion of outlier results during study selection [15]. Consequently, the true prevalence of the OSSN could actually be higher than shown by our results. Another important limitation of the current evidence is the lack of reported clinical outcome data, such as pain levels, functional impairment, as well as the scarcity of laterality information and morphometric measurements. Furthermore, most included studies relied on retrospective imaging reviews, which may introduce selection and reporting biases. We encourage future research to address these gaps to enable a more comprehensive and clinically applicable understanding of the OSSN in practice.

## 5. Conclusions

Our study demonstrates that the OSSN is an uncommon accessory bone, typically asymptomatic, though in rare instances its presence can result in a painful syndrome or predispose individuals to NSFs. While our findings suggest a slightly higher prevalence in males and in the European population, these differences were not statistically significant. Accurate identification of the OSSN and its differentiation from avulsion fractures are essential, as misinterpretation may lead to unwarranted and potentially invasive interventions. Physicians should consider this ossicle in the differential diagnosis of otherwise unexplained dorsal foot pain and manage it with appropriate imaging and a stepwise approach from conservative care to, if needed, surgical intervention.

## Figures and Tables

**Figure 1 jcm-14-05934-f001:**
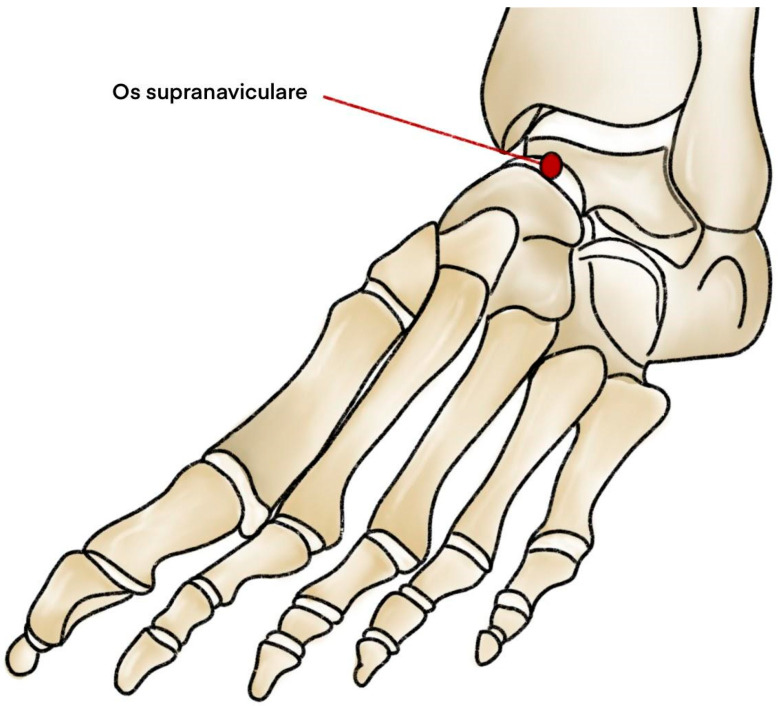
Diagram illustrating the anatomical location of the OSSN.

**Figure 2 jcm-14-05934-f002:**
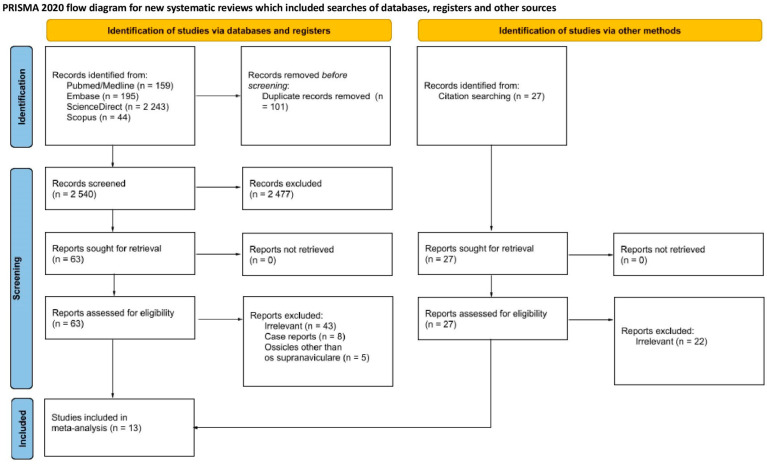
PRISMA flowchart illustrating the study selection process. PROSPERO (ID: CRD42025638111).

**Figure 3 jcm-14-05934-f003:**
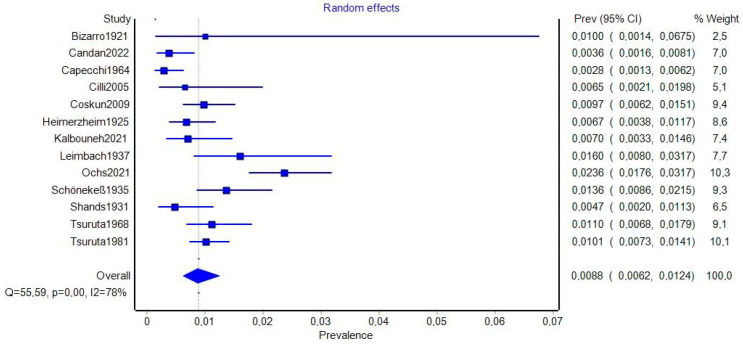
Forest plot illustrating the general prevalence estimate of OSSN across included studies [1,3,17,18,19,20,21,22,23,24,25,26,27].

**Figure 4 jcm-14-05934-f004:**
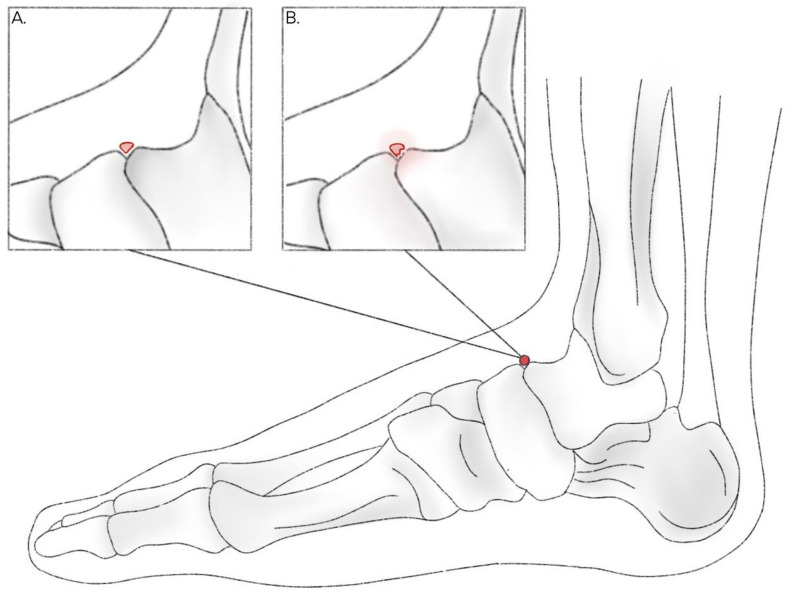
Visual comparison between (**A**) Os supranaviculare and (**B**) navicular avulsion fracture.

**Table 1 jcm-14-05934-t001:** Characteristics of included studies.

Author, Year	Origin of the Study	Diagnostic Modality	Prevalence Reporting Format (per Foot/per Patient)	Number of Feet Examined	Number of Feet with OSSN
Bizarro, 1921 [19]	England	X-ray	Per foot	100	1
Candan, 2022 [3]	Turkey	X-ray	Per foot	1651	6
Capecchi, 1964 [20]	Italy	X-ray	Per foot	2155	6
Cilli, 2005 [18]	Turkey	X-ray	Per foot	464	3
Coskun, 2009 [1]	Turkey	X-ray	Per foot	1968	19
Heimerzheim, 1925 [21]	England	X-ray	Per foot	1800	12
Kalbouneh, 2021 [27]	Jordan	X-ray	Per foot	1000	7
Leimbah, 1937 [22]	Germany	X-ray	Per foot	500	8
Ochs, 2021 [26]	Germany	X-ray	Per foot	1820	43
Schönekeß, 1935 [23]	Germany	X-ray	Per foot	1324	18
Shands, 1931 [25]	USA	X-ray	Per foot	1054	5
Tsuruta, 1968 [24]	Japan	X-ray	Per foot	1449	16
Tsuruta, 1981 [17]	Japan	X-ray	Per foot	3460	35

Abbreviations: OSSN, os supranaviculare.

**Table 2 jcm-14-05934-t002:** Results of the quality assessment of included studies.

Risk of Bias					
Author, Year	Objective(S) and Study Characteristics	Study Design	Methodology Characterization	Descriptive Anatomy	Reporting of Results
Bizarro, 1921 [19]	LOW	UNCLEAR	LOW	LOW	LOW
Candan, 2022 [3]	LOW	LOW	LOW	LOW	LOW
Capecchi, 1964 [20]	LOW	HIGH	HIGH	LOW	LOW
Cilli, 2005 [18]	LOW	LOW	LOW	LOW	LOW
Coskun, 2009 [1]	LOW	LOW	LOW	LOW	LOW
Heimerzheim, 1925 [21]	UNCLEAR	LOW	UNCLEAR	UNCLEAR	LOW
Kalbouneh, 2021 [27]	LOW	LOW	LOW	LOW	LOW
Leimbah, 1937 [22]	LOW	LOW	UNCLEAR	UNCLEAR	LOW
Ochs, 2021 [26]	LOW	LOW	LOW	LOW	LOW
Schönekeß, 1935 [23]	LOW	UNCLEAR	LOW	LOW	LOW
Shands, 1931 [25]	UNCLEAR	LOW	LOW	LOW	LOW
Tsuruta, 1968 [24]	LOW	LOW	UNCLEAR	LOW	LOW
Tsuruta, 1981 [17]	LOW	LOW	UNCLEAR	LOW	LOW

**Table 3 jcm-14-05934-t003:** Prevalence of OSSN in different subgroups.

Category	Subgroup	Number of Studies (Number of Feet)	Number of OSSN	Prevalence (95%CI); (95%PI)	*I*^2^ (95%CI)	Cochran’s Q; *p*-Value
General		13 (18,745)	179	0.88% (0.62–1.24%); (0.26–2.97%)	78.41% (63.59–87.20%)	55.59; *p* < 0.001
Sex	Males	3 (2986)	25	0.87% (0.58–1.32%); -	7.75% (0.00–90.40%)	2.17; *p* = 0.338
	Females	3 (3125)	23	0.48% (0.14–1.64%); -	68.80% (0.00–90.93%)	6.41; *p* = 0.041
Origin	Asia	6 (9992)	86	0.87% (0.66–1.13%); (0.47–1.61%)	28.58% (0.00–70.65%)	7.00; *p* = 0.221
	Europe	6 (7699)	88	1.04% (0.55–1.96%); (0.12–8.32%)	85.53% (70.47–92.91%)	34.55; *p* < 0.001
Sample size	Insufficient (<1521 feet)	7 (5891)	58	1.02% (0.75–1.39%); (0.54–1.94%)	22.58% (0.00–65.51%)	7.75; *p* = 0.257
	Sufficient (>1521 feet)	6 (12,854)	121	0.79% (0.44–1.42%); (0.10–6.02%)	89.48% (79.76–94.53%)	47.52; *p* < 0.001

Abbreviations: OSSN, os supranaviculare; 95%CI, 95% confidence intervals; 95%PI, 95% prediction intervals.

## Data Availability

Dataset available on request from the authors.

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
