# Peer review of "Prevalence of the Os Supranaviculare: A Systematic Review with Meta-Analysis"

_jcm, 2025, doi:10.3390/jcm14175934_

Round 1
Reviewer 1 Report
Comments and Suggestions for Authors
This study is a meta-analysis of 13 articles investigating the epidemiology of os supranaviculare prevalence in the general population. The authors estimate an overall prevalence of approximately 1%, slightly higher in males, although this difference is not statistically significant. No major differences were observed between studies conducted on European and Asian populations.
The authors acknowledge the main limitations of the study, particularly the heterogeneity of the included studies and the inability to determine the proportion of cases in which the os supranaviculare is symptomatic. Nonetheless, the study’s methodological rigor and comprehensive statistical analysis make it a valuable and informative contribution.
Just two minor suggestions for improvement:
-
Consider adding radiographic images, if available, showing both the os supranaviculare and avulsion fractures of the navicular bone. This would enhance both the clarity and visual appeal of the article for readers.
-
If feasible, given the systematic nature of the review and the use of specific keywords, it would be helpful to include the number of published reports (including case reports and case series) describing asymptomatic os supranaviculare, symptomatic cases, and surgically treated cases. Although such reports are likely scarce, this addition would help further delineate and contextualize the current state of knowledge on the topic.
Author Response
Comments 1: Consider adding radiographic images, if available, showing both the os supranaviculare and avulsion fractures of the navicular bone. This would enhance both the clarity and visual appeal of the article for readers.
Response 1: Dear Reviewer, we sincerely appreciate this thoughtful suggestion. We fully agree that illustrating the differences between os supranaviculare and avulsion fractures can further enhance the clarity and visual appeal of the manuscript. Following this recommendation, we carefully reviewed all accessible radiographs from our clinic; however, we were unfortunately unable to identify images of sufficient quality to allow a meaningful side-by-side comparison of these two entities. We regret not being able to include such radiographs. We are truly grateful for your insightful comment, which encouraged us to carefully consider the visual presentation of these entities, and we hope that our current graphical representation still effectively supports readers’ understanding of the condition.
Comments 2: If feasible, given the systematic nature of the review and the use of specific keywords, it would be helpful to include the number of published reports (including case reports and case series) describing asymptomatic os supranaviculare, symptomatic cases, and surgically treated cases. Although such reports are likely scarce, this addition would help further delineate and contextualize the current state of knowledge on the topic.
Response 2: We sincerely thank the reviewer for the insightful suggestion to provide additional context regarding published reports of asymptomatic, symptomatic, and surgically treated OSSN. In our systematic search, we identified several case reports and case series, most of which focused on symptomatic OSSN, particularly in athletes where repetitive midfoot loading appears to play a key role. Reports of asymptomatic OSSN are less frequent, and surgically treated cases are rare. To better contextualize the current state of knowledge, we have now included a brief summary of these findings in the Discussion section. We believe that this addition provides a clearer perspective on the clinical spectrum and research landscape of OSSN.
(page 11, verses 355-367)
Reviewer 2 Report
Comments and Suggestions for Authors
The work is comprehensive and thoroughly done, with a very interesting topic and a clinically significant contribution.
Line 20: Please, put the comma after “clinical significance” ....
Line 49: “especially common” replace with “widespread”
Line 68: “OS” – OSSN
Line 76: “Pubmed” – PubMed
Line 78: The following search terms…
Line 109: Please, explain the AQUA abbreviation.
Line 125: .... of THE applied transformation ....
Line 143: ... strongly suggest THE presence; … IMPACT instead of effect.
Line 173: “to be at” - AS BEEING “low”
Line 176: PRIMARILY instead of largely
Line 190: “OS” – OSSN
Line 222: Please, put the COMMA after the “geographic region”
Line 224: “variant” – variation
Line 278: ... seen on A lateral ankle radiograph
Line 292: According to Terminologia anatomica from 2019. deep peroneal nerve is English synonym for DEEP FIBULAR NERVE. Deep fibular nerve is prefered term.
Line 296: “occur often” – often occur
Line 298: USUALLY, instead of often.
Line 320: put the COMMA after South America.
Line 322: “suggest” – suggests
Author Response
Comments 1: The work is comprehensive and thoroughly done, with a very interesting topic and a clinically significant contribution.
Line 20: Please, put the comma after “clinical significance” ....
Line 49: “especially common” replace with “widespread”
Line 68: “OS” – OSSN
Line 76: “Pubmed” – PubMed
Line 78: The following search terms…
Line 109: Please, explain the AQUA abbreviation.
Line 125: .... of THE applied transformation ....
Line 143: ... strongly suggest THE presence; … IMPACT instead of effect.
Line 173: “to be at” - AS BEEING “low”
Line 176: PRIMARILY instead of largely
Line 190: “OS” – OSSN
Line 222: Please, put the COMMA after the “geographic region”
Line 224: “variant” – variation
Line 278: ... seen on A lateral ankle radiograph
Line 292: According to Terminologia anatomica from 2019. deep peroneal nerve is English synonym for DEEP FIBULAR NERVE. Deep fibular nerve is prefered term.
Line 296: “occur often” – often occur
Line 298: USUALLY, instead of often.
Line 320: put the COMMA after South America.
Line 322: “suggest” – suggests
Response 1: We sincerely thank the reviewer for the positive feedback and for the meticulous, line-by-line suggestions to improve clarity, precision, and adherence to correct terminology. We have carefully implemented all the recommended changes, including adjustments to punctuation, word choice, terminology consistency, abbreviation expansions, and anatomical nomenclature, ensuring the manuscript meets high standards of readability and accuracy.
Reviewer 3 Report
Comments and Suggestions for Authors
The manuscript addresses an important anatomical variant with potential clinical relevance, particularly in orthopedics and radiology. Overall, the study is methodologically robust, well-structured, and contributes meaningful pooled data on a scarcely studied ossicle. Nevertheless, several aspects of the manuscript could benefit from improvement to enhance its clarity, transparency, and clinical applicability.
The introduction offers a good general overview of Os supranaviculare (OSSN), its anatomical location, terminology, and clinical relevance. It also introduces the problem of inconsistent prevalence data across the literature. However, it could be improved by further elaborating on the biomechanical implications, differential diagnosis challenges, and clinical consequences of misinterpreting OSSN as an avulsion fracture, which are only briefly mentioned. Additionally, while many foundational references are cited, the section lacks recent and comprehensive clinical literature that would better link the anatomical variant to contemporary diagnostic and management protocols, especially in orthopedics and sports medicine. Including such references and contextualizing the ossicle's relevance within modern imaging and treatment practices would strengthen the background and better justify the study's aims.
The methodology is prepare in a structured and generally comprehensive manner, including database searches, selection criteria, data extraction, risk of bias assessment (using the AQUA tool), and statistical analyses. However, some key methodological details are only available in the Supplementary Materials, such as the exact search terms, the results of the AQUA quality assessment per study, and the specific data extraction template. Including summaries or highlights of these components within the main text would enhance transparency and reproducibility. Furthermore, the authors could have discussed in more depth the inter-rater reliability for study selection and bias assessment (e.g., Cohen’s kappa) and elaborated on the management of missing data or language barriers, which is only briefly mentioned.
The results are logically organized and supported by relevant tables and forest plots. However, the presentation could be improved in several ways. First, there is limited reporting of laterality and morphometric data regarding the OSSN, which is acknowledged by the authors but could have been visually represented to emphasize these gaps in the literature. Second, the heterogeneity of the studies is substantial (I² > 70% in multiple subgroups), but the presentation does not fully explore or visualize these discrepancies. The inclusion of more stratified plots, geographical mapping, or sensitivity analysis visuals could significantly enhance reader comprehension. Lastly, the absence of clinical outcome data (e.g., pain, treatment) limits the interpretability of the findings, particularly since the title emphasizes “clinical implications.”
The manuscript contains essential and well-structured figures, including a PRISMA flow diagram, forest plot of pooled prevalence, and radiographic comparisons. The tables summarizing study characteristics and subgroup analyses are also clear. However, the AQUA quality assessments, which are central to evaluating study reliability, are not summarized in a table within the main text but relegated to Supplementary Material 3. Including a concise table with the risk of bias per domain per study would strengthen methodological transparency. Additionally, no visual summary (e.g., a map or bar chart) is provided to illustrate the geographical distribution of OSSN prevalence, which could help readers better understand regional trends. Enhancing figure diversity and integrating more graphical summaries of key outcomes would improve the overall clarity and accessibility of the results.
In conclusion, while the manuscript is generally well-conceived and provides valuable insights into the prevalence of Os supranaviculare, addressing the aforementioned areas for improvement, particularly regarding the depth of background information, presentation of results, inclusion of key methodological details in the main text, and the enhancement of figures, would strengthen the paper’s overall impact and clarity
Author Response
Comments 1: The introduction offers a good general overview of Os supranaviculare (OSSN), its anatomical location, terminology, and clinical relevance. It also introduces the problem of inconsistent prevalence data across the literature. However, it could be improved by further elaborating on the biomechanical implications, differential diagnosis challenges, and clinical consequences of misinterpreting OSSN as an avulsion fracture, which are only briefly mentioned. Additionally, while many foundational references are cited, the section lacks recent and comprehensive clinical literature that would better link the anatomical variant to contemporary diagnostic and management protocols, especially in orthopedics and sports medicine. Including such references and contextualizing the ossicle's relevance within modern imaging and treatment practices would strengthen the background and better justify the study's aims.
Response 1: We sincerely thank the Reviewer for this perceptive and constructive suggestion. We have revised the Introduction to briefly expand on the biomechanical implications of OSSN, clarify the differential-diagnosis challenges versus avulsion fractures, and emphasise the potential clinical consequences of misclassification (including differing management pathways). We also added several recent, clinically relevant references that better connect the anatomical variant to contemporary imaging and treatment practices. We are grateful for this guidance, which materially improved the manuscript’s clinical framing and rationale.
(page 2, verses 49-66)
Comments 2: The methodology is prepare in a structured and generally comprehensive manner, including database searches, selection criteria, data extraction, risk of bias assessment (using the AQUA tool), and statistical analyses. However, some key methodological details are only available in the Supplementary Materials, such as the exact search terms, the results of the AQUA quality assessment per study, and the specific data extraction template. Including summaries or highlights of these components within the main text would enhance transparency and reproducibility. Furthermore, the authors could have discussed in more depth the inter-rater reliability for study selection and bias assessment (e.g., Cohen’s kappa) and elaborated on the management of missing data or language barriers, which is only briefly mentioned.
Response 2: Dear Reviewer, thank you for the valuable feedback regarding the methodological reporting. In response, we have expanded the Methods section to enhance transparency and reproducibility. We have elaborated on the exact search terms used across the databases and clarified procedures for handling non-English publications and unclear data. Additionally, we included a brief description of the pre-defined data extraction template, outlining the variables collected per study to improve clarity for readers. The results of the AQUA quality assessment, previously provided only in Supplementary Material 3, have now been moved to Table 2 for easier reference. To further strengthen the methodological rigor, we added Cohen’s kappa to quantify inter-rater agreement for study selection and bias assessment, and we clarified that missing data were handled using the pairwise deletion method. We believe these additions substantially address the reviewer’s concerns and improve both the transparency and reproducibility of our study.
(Table 2; page 3 verses 90-98; page 4 verses 123-131; page 5 verses 160-161; page 4 verses 114-119)
Comments 3: The results are logically organized and supported by relevant tables and forest plots. However, the presentation could be improved in several ways. First, there is limited reporting of laterality and morphometric data regarding the OSSN, which is acknowledged by the authors but could have been visually represented to emphasize these gaps in the literature. Second, the heterogeneity of the studies is substantial (I² > 70% in multiple subgroups), but the presentation does not fully explore or visualize these discrepancies. The inclusion of more stratified plots, geographical mapping, or sensitivity analysis visuals could significantly enhance reader comprehension. Lastly, the absence of clinical outcome data (e.g., pain, treatment) limits the interpretability of the findings, particularly since the title emphasizes “clinical implications.”
Response 3: We sincerely thank the Reviewer for these thoughtful and constructive observations, which we believe will substantially improve the clarity and visual impact of our manuscript. We fully agree that additional visual representations will enhance the reader’s understanding, particularly in light of the observed heterogeneity. In the revised version, we have therefore added stratified plots to display subgroup variations, allowing readers to appreciate potential geographical trends. We have also revised our Discussion section and clearly state the limitations of our study, and how the absence of the clinical outcome data impacts the interpretability of our findings. Furthermore, in accordance with another Reviewer’s valuable suggestion, we have revised the manuscript title to more accurately reflect the available data. We believe these changes will strengthen both the clarity and interpretability of our work. We believe these additions address the Reviewer’s suggestions and strengthen the manuscript’s overall presentation and interpretability.
(Title; Supplementary Material 4; page 12, verses 387-393)
Comments 4: The manuscript contains essential and well-structured figures, including a PRISMA flow diagram, forest plot of pooled prevalence, and radiographic comparisons. The tables summarizing study characteristics and subgroup analyses are also clear. However, the AQUA quality assessments, which are central to evaluating study reliability, are not summarized in a table within the main text but relegated to Supplementary Material 3. Including a concise table with the risk of bias per domain per study would strengthen methodological transparency. Additionally, no visual summary (e.g., a map or bar chart) is provided to illustrate the geographical distribution of OSSN prevalence, which could help readers better understand regional trends. Enhancing figure diversity and integrating more graphical summaries of key outcomes would improve the overall clarity and accessibility of the results.
Response 4: We thank the reviewer for acknowledging the clarity and structure of our figures and tables. In response to the concern regarding the AQUA quality assessments, we have moved the results previously provided in Supplementary Material 3 into Table 2 in the main text. This table now summarizes the risk of bias per domain for each included study, allowing readers to directly evaluate study reliability without referring to supplementary materials. Regarding the visual presentation of geographical prevalence, we recognize the value of such summaries. While we were unable to create a geographical map, we have provided stratified plots that clearly illustrate differences in prevalence across regions and study characteristics. These plots enable readers to visually appreciate variations and enhance the overall accessibility and interpretability of the results. We believe that these updates significantly improve methodological transparency and the accessibility of our results, aligning with the reviewer’s suggestions to enhance figure diversity and provide graphical summaries of key outcomes.
(Table 2; Supplementary Material 4)
Comments 5: In conclusion, while the manuscript is generally well-conceived and provides valuable insights into the prevalence of Os supranaviculare, addressing the aforementioned areas for improvement, particularly regarding the depth of background information, presentation of results, inclusion of key methodological details in the main text, and the enhancement of figures, would strengthen the paper’s overall impact and clarity
Response 5: We sincerely thank the reviewer for their encouraging and constructive feedback. We greatly appreciate the recognition of the value of our work and agree that the suggested refinements would strengthen the manuscript. We have carefully considered your remarks and taken steps to enhance the depth of the background, improve the clarity of the result presentation, and ensure key methodological details are clearly included in the main text. Additionally, we have enhanced the figures to provide a more intuitive and accessible visualization of our findings. We believe these improvements substantially increase the clarity, readability, and overall impact of the manuscript, and we are grateful for your thoughtful guidance.
Reviewer 4 Report
Comments and Suggestions for Authors
Prevalence and clinical implications of os supranaviculare: A systematic review with meta-analysis
The title of the manuscript should be revised, as in this systematic review clinical implications of Os Supranaviculare were not studied either in the search strategy or in the inclusion criteria.
This comment applies to all relevant sections of the manuscript, including the abstract, conclusion, introduction section (the last paragraph), and main conclusion of the study.
In the introduction, in the last paragraph, it has been stated that “This meta-analysis aims to clarify and systematically summarize all available data on the anatomical characteristics and clinical significance. However, the information on anatomical characteristics and clinical significance has not been added in the results section.
In the methodology, the inclusion and exclusion criteria should be revised to make it clear to the reader.
The PROSPERO registered number should be added in the legend of figure 2.
In the results section, the data on anatomical characteristics and symptoms associated with them should be analyzed and added in detail.
All other sections of results are presented well.
The discussion section is written overall well. The first part of this section should be divided into small paragraphs.
The limitations of the study should be added.
Author Response
Comments 1: The title of the manuscript should be revised, as in this systematic review clinical implications of Os Supranaviculare were not studied either in the search strategy or in the inclusion criteria. This comment applies to all relevant sections of the manuscript, including the abstract, conclusion, introduction section (the last paragraph), and main conclusion of the study.
Response 1: We sincerely thank the Reviewer for this important observation. We agree that the current title may suggest a focus on clinical implications, which were not directly assessed in our search strategy or inclusion criteria. In accordance with this valuable suggestion, we have revised the title to more accurately reflect the scope and content of our systematic review. We believe this revision will provide greater clarity and alignment between the study’s objectives and its presentation.
Comments 2: In the introduction, in the last paragraph, it has been stated that “This meta-analysis aims to clarify and systematically summarize all available data on the anatomical characteristics and clinical significance. However, the information on anatomical characteristics and clinical significance has not been added in the results section.
Response 2: Dear Reviewer, thank you for pointing out this inconsistency between our stated aims in the introduction and the content presented in the results. We agree that the results section did not explicitly present data on anatomical characteristics and clinical significance as indicated. In the revised manuscript, we have updated the introduction’s final paragraph to ensure that the study aims are aligned with the scope of the data collected and reported.
(page 2, verses 72-74)
Comments 3: In the methodology, the inclusion and exclusion criteria should be revised to make it clear to the reader.
Response 3: We sincerely thank the Reviewer for this helpful suggestion. We agree that clarifying the inclusion and exclusion criteria will improve the transparency and readability of the methodology. In the revised manuscript, we have rewritten this section to present the criteria in a clearer and more structured format, specifying each point separately to ensure that readers can easily understand the study selection process.
(pages 3-4, verses 99-118)
Comments 4: The PROSPERO registered number should be added in the legend of figure 2.
Response 4: We thank the Reviewer for this valuable observation. We have added the PROSPERO registration number to the legend of Figure 2 in the revised manuscript to ensure complete reporting and transparency.
(page 6, verses 192-193)
Comments 5: In the results section, the data on anatomical characteristics and symptoms associated with them should be analyzed and added in detail.
Response 5: Dear Reviewer, we sincerely appreciate this valuable suggestion. We agree that a more detailed presentation of anatomical characteristics and any associated symptoms would enhance clarity. While such data were already present in the results section, we have now revised and expanded this subsection to explicitly highlight available anatomical findings, including laterality, and to clearly state that morphometric and symptom-related data was not sufficient for analysis. We believe these changes improve readability and better convey the scope of the available evidence.
(page 8, verses 226-230)
Comments 6: All other sections of results are presented well. The discussion section is written overall well. The first part of this section should be divided into small paragraphs.
Response 6: We thank the Reviewer for the positive feedback on the presentation of the results and discussion. We appreciate the suggestion to divide the first part of the discussion into smaller paragraphs to enhance readability. In the revised manuscript, we have reorganized this section into shorter, thematically focused paragraphs while preserving the original content, to improve clarity and flow for the reader.
(pages 10-11, verses 262-305)
Comments 7: The limitations of the study should be added.
Response 7: We thank the Reviewer for this important suggestion. In the revised manuscript, we have renamed the “Methodological considerations” section to “Methodological considerations and limitations” and expanded it to explicitly outline the study’s limitations. In addition to the points already discussed (substantial heterogeneity, limited geographic representation, potential publication bias), we have added the lack of clinical outcome, scarcity of morphometric and laterality information and the predominance of retrospective imaging reviews as further limitations. We believe these additions improve the transparency of our work and help readers better contextualize our findings.
(page 12, verses 370-392)